# Converging Role for REEP1/SPG31 in Oxidative Stress

**DOI:** 10.3390/ijms24043527

**Published:** 2023-02-09

**Authors:** Valentina Naef, Maria C. Meschini, Alessandra Tessa, Federica Morani, Debora Corsinovi, Asahi Ogi, Maria Marchese, Michela Ori, Filippo M. Santorelli, Stefano Doccini

**Affiliations:** 1Molecular Medicine for Neurodegenerative and Neuromuscular Diseases Unit, IRCCS Stella Maris Foundation, 56128 Pisa, Italy; 2Department of Biology, University of Pisa, 56126 Pisa, Italy; 3Department of Translational Research and New Technologies in Medicine and Surgery, University of Pisa, 56126 Pisa, Italy

**Keywords:** mitochondria, REEP1, SPG31, bioenergetic defects, ROS, zebrafish, resveratrol

## Abstract

Mutations in the receptor expression-enhancing protein 1 gene *(REEP1)* are associated with hereditary spastic paraplegia type 31 (SPG31), a neurological disorder characterized by length-dependent degeneration of upper motor neuron axons. Mitochondrial dysfunctions have been observed in patients harboring pathogenic variants in *REEP1*, suggesting a key role of bioenergetics in disease-related manifestations. Nevertheless, the regulation of mitochondrial function in SPG31 remains unclear. To elucidate the pathophysiology underlying REEP1 deficiency, we analyzed in vitro the impact of two different mutations on mitochondrial metabolism. Together with mitochondrial morphology abnormalities, loss-of-*REEP1* expression highlighted a reduced ATP production with increased susceptibility to oxidative stress. Furthermore, to translate these findings from in vitro to preclinical models, we knocked down *REEP1* in zebrafish. Zebrafish larvae showed a significant defect in motor axon outgrowth leading to motor impairment, mitochondrial dysfunction, and reactive oxygen species accumulation. Protective antioxidant agents such as resveratrol rescued free radical overproduction and ameliorated the SPG31 phenotype both in vitro and in vivo. Together, our findings offer new opportunities to counteract neurodegeneration in SPG31.

## 1. Introduction

Hereditary spastic paraplegia (HSP) is a group of clinically and genetically heterogeneous neurodegenerative diseases characterized by corticospinal tract lesions, which may be combined with additional neurological or non-neurological manifestations [1]. To date, about 80 distinct spastic gait disease loci have been identified, and the corresponding gene products have been linked to a limited number of molecular mechanisms, including lipid metabolism, axonal transport, protein folding, endosome membrane trafficking, autophagy, endoplasmic reticulum (ER) membrane modeling, and oxidative phosphorylation metabolism [1,2,3].

Pathogenic variants in the gene encoding mitochondrial receptor expression-enhancing protein 1 (*REEP1*) are associated with HSP type 31 (SPG31; MIM 610250), the third most common cause of autosomal dominant HSP (AD-HSP), accounting for about 5% of young patients [4]. Although the precise steps through which mutations in *REEP1* lead to HSP are still poorly understood, loss of function and haploinsufficiency are considered major mechanisms of action in SPG31 [4,5,6,7,8]. REEP1 protein controls the shape/dynamics of different endomembranes through a microtubule-associated role, and multiple studies highlighted its involvement in ER formation and curving, in the trafficking of mitochondria, and in lipid droplet biology [9,10,11].

At the subcellular level, REEP1 seems to act at the contact between ER and mitochondria (mitochondria-associated membrane, MAMs) [10], which represent the key site of location or recruitment of many proteins involved in the regulation and maintenance of mitochondrial morphology and dynamics (e.g., the mitochondrial fission protein DRP1 and the mitochondrial fusion protein MFN2) [12,13,14]. In line with this, mitochondrial dysfunctions, such as reduced mitochondrial oxidative metabolism in skeletal muscle and abnormal mitochondrial network organization in skin fibroblasts, have been observed in an SPG31 patient [6], suggesting a potential relationship between *REEP1* mutations, energy production, and mitochondrial dynamics. Interestingly, similar alterations have already been reported in other forms of HSP, including SPG7 and SPG13, which are caused by mutations in the mitochondrial proteins Paraplegin and Heat-shock protein 60, respectively [15,16,17]. The non-mitochondrial REEP1-interaction-partner Spastin (SPG4) has also been shown to strongly affect mitochondrial morphology through the regulation of microtubule dynamics [18,19].

Oxidative stress is central to various neurodegenerative disorders since neurons are particularly vulnerable to reactive oxygen species (ROS). Redox imbalance is closely connected with mitochondria as both source and target of ROS, affecting their bioenergetic capacity and metabolic functions. An increased sensitivity towards oxidative damage, which is accompanied by depolarization of mitochondria, has been observed as a consequence of reduced axonal transport HSP-related [20], indicating the influence of oxidative damage in the degenerative process of HSP. Moreover, plasma levels of superoxide dismutase, vitamin E, and nitric oxide were significantly reduced in a heterogeneous cohort of HSP patients suggesting inhibition of free radical scavenging activity [21]. Previous studies have indicated that antioxidants often achieve their neuroprotective activity by lowering ROS contents in cells; among others, resveratrol has shown great potential by directly scavenging overproduced ROS [22].

In order to investigate the impact of mitochondrial metabolism and dynamic in the pathophysiology of REEP1 deficiency and analyze the interplay between mitochondrial impairment and oxidative stress in SPG31. We collected primary fibroblasts from four patients in three unrelated SPG31 families, demonstrating how the loss-of-*REEP1* expression is accompanied by reduced ATP production with increased susceptibility to oxidative stress. Moreover, to translate these findings from in vitro to preclinical models, we characterized a *reep1* knockdown (KD) zebrafish model. Zebrafish is a tool for studying HSP [23,24,25,26], and it represents a valid alternative to the use of more common animal models, such as mice [27,28], specifically for the study of early neurodevelopment. We showed that *reep1* morphants displayed a dramatic defect in motor axon outgrowth leading to motor impairment, mitochondrial dysfunction, and ROS accumulation. Alongside cells from *REEP1* patients, morphants underwent pharmacological treatments to modulate the susceptibility to oxidative stress in SPG31.

## 2. Results

### 2.1. Molecular Analysis in SPG31/REEP1 Cells and Zebrafish

We identified two heterozygous pathogenic variants in *REEP1* (NM_001371279.1) in four patients from three unrelated families. The Appendix A reports the main clinical features of analyzed patients. The proband in Family 1 harbored the novel [c.401_409dup; c.410_417 + 11del] (p.?) variant (hereinafter referred in figures to as Variant 1) located at the edge exon 5-intron 5; the probands in the unrelated families 2 and 3 harbored the already reported c. 337C > T (p.Arg113*) [29] (Variant 2 in figures), predicted to cleave 89 amino acids at the C-terminus of the protein (Figure 1A). Sequence analysis of the cDNA of *REEP1* did not reveal the pathogenic variants in mRNA from skin fibroblasts of index cases in the three families, suggesting that mutant transcripts were either subject to degradation by the nonsense-mediated mRNA decay or not formed. REEP1 expression levels by Western blotting (WB) showed reduced protein levels (Figure 1B). 

To examine the spatio-temporal expression of *reep1* mRNA during zebrafish embryogenesis, whole-mount in situ hybridization (WISH) was performed in wild-type (WT) zebrafish embryos from the stage 24 h post fertilization (hpf) to 120 hpf. At 24 hpf, *reep1* was mainly expressed in the myotome. At later developmental stages, the transcript was closely distributed in the most anterior region of the central nervous system, with weaker expression levels detected in the trunk and tail. qRT-PCR analysis between 24 and 120 hpf revealed that *reep1* was zygotically expressed since its transcript was detected starting from 24 hpf with a peak at 72 hpf (Figure 1C).

### 2.2. Functional and Morphological Studies in REEP1 Mutated Fibroblasts

To analyze the direct consequence of specific *REEP1* alterations in the mitochondrial compartment, we investigated mutant skin fibroblasts in real-time by micro-oxygraphy using Seahorse XFe24 Analyzer (Agilent, Santa Clara, CA, USA). Compared to control cells, Oxygen Consumption Rate (OCR) was severely impaired with a significant reduction of bioenergetic parameters such as ATP production and Spare Respiratory Capacity (Figure 2A). To study the impact of impaired bioenergetic functions on mitochondrial membrane potential (∆ψ_m_), we evaluated the steady-state loading of the potentiometric dye tetramethylrhodamine methyl ester (TMRM) and observed a reduced dye accumulation in mutant cells when compared to control skin fibroblasts (Figure 2B). Moreover, we reasoned that a latent mitochondrial dysfunction in *REEP1*-cells could also be masked by ATP synthase operating in the reverse mode toward hydrolysis. To test this hypothesis, we measured ∆ψ_m_ by real-time imaging of TMRM fluorescence intensity after incubation with oligomycin, an inhibitor of mitochondrial F_1_F_O_-ATP synthase. The TMRM fluorescence profiles of mutant fibroblasts confirmed a low basal signal, even though after the addition of the inhibitor, mutated cells were able to maintain their ∆ψ_m_ (Figure 2C)_,_ in contrast to what has previously been described in respiratory chain complex I deficiency [30]. Following the oligomycin-measurements, FCCP was added to collapse ∆ψ_m_ and resulted in a rapid dissipation of the TMRM signal. Similar fluorescence values after FCCP addition indicate a comparable penetration of TMRM-dye and therefore an adequate sensitivity to record changes in ∆ψ_m_.

Alterations of mitochondrial morphology have been previously described in different forms of HSP. In particular, REEP1 dysfunction in SPG31 directly affected mitochondrial dynamics by the inhibition of mitochondrial fission protein, DRP1, due to the hyper-phosphorylation of its serine 637 residue [11]. A reduced expression of DRP1 was observed by WB also in our cells (Figure 3A), whereas the steady-state levels of OPA1, MNF2, and FIS1, additional proteins involved in mitochondrial dynamics, were similar to those observed in control fibroblasts (see Appendix A). To evaluate the fusion activity of DRP1, DRP1–Ser-637 phosphorylation was triggered by the cAMP-dependent protein kinase A (PKA) activation via 24-h starvation treatment in the EBSS medium. An unbalanced ratio between total DRP1 and its Ser 637-phosphorilated form was observed in both mutant cell lines demonstrating a specific DRP1 hyperphosphorylation associated with REEP1 defect (Figure 3B).

To further characterize the cellular phenotype associated with *REEP1* mutations, we measured the morphometric parameters of mitochondrial network aspect ratio (AR) and form factor (FF) as a function of mitochondrial length and branching, respectively. The analysis showed a hyperfused network with increased percentages of tubular mitochondria, together with a higher degree of branching and elongation (Figure 4A,B). These features reinforce the notion that REEP1 is involved in fission processes [6,11], and recall the hyperfused mitochondrial network previously described in a patient harboring a dominant-negative mutation in the dynamin-like protein 1 gene [31] and in *DRP1* KO models [32].

To clarify the relationship between mitochondrial depolarization and network shape, SPG31 cultured skin fibroblasts labeled with MitoTracker Red were incubated with 2μM of carbonyl cyanide-4-(trifluoromethoxy) phenylhydrazone (FCCP) for 30 min to induce fragmentation of the mitochondrial network. Contrary to normal cells, under FCCP treatment, the *REEP1* mutated fibroblasts resist the fragmentation *stimulus*, confirming that DRP1 is required for fragmentation induced by mitochondrial depolarization [33] and that the hyperfused network in SPG31 cells depends on DPR1 hyperphosphorylation (Figure 4C). Taken together, these data in skin cells indicate that REEP1 regulates the morphology of the mitochondrial network and is essential for energy production and homeostasis.

### 2.3. Characterization of the Zebrafish Reep1 KD Model

To investigate how *reep1* functions during zebrafish development, we generated a KD morphant model using a specific morpholino (MO ATG *reep1*) and assessed the locomotor behavior of morphant larvae at different stages of development. Analysis of tail flicks at 30 hpf in *reep1* morphant embryos showed a significant decrease in burst activity (i.e., the percentage of time an embryo moving) compared to controls (Figure 5A). Touch-response test at 48 hpf (Figure 5B) showed a marked reduction of distance and velocity in morphants when compared to control embryos (Figure 5C,C’) (see Video S1). We assessed whether this loss of mobility was specifically associated with the lack of *reep1* using rescue experiments. This was performed by co-injections of human *REEP1* mRNA with *reep1* morpholino in one-cell stage embryos and analyzing the locomotor behavior of the double-injected embryos at 24 hpf and 48 hpf (Figure 5A,B). We observed that the motility of the morphant embryos was fully rescued by the expression of human wild-type mRNA (Figure 5A,B), corroborating the impression that the swimming deficit was caused by *reep1* KD. At 120 hpf, video tracking data revealed a significantly reduced locomotor activity of *reep1* morphant larvae in terms of both velocity and distance covered (Figure 5C,C’). To explore the nature of this motor impairment, we carried out immunolabeling of the spinal neuron axonal tracts of morphant and control embryos using the motor axon marker znp1 and acetylated alfa-tubulin, a pan-neuronal axonal marker (Figure 5D,E’). Immunostaining in 48 hpf morphants with acetylated alfa-tubulin showed abnormal axon pathfinding of spinal motor neurons (SMNs) with multiple aberrant branching all along the axons (Figure 5D,D’). Moreover, labeling of the same morphants siblings at 120 hpf for znp1 showed that SMN axons were less defined and thinner than controls (Figure 5E,E’). This indicated that the distal region of morphant SMN axons was abnormally developed. To further characterize the model, we performed bioenergetic assays in morphants. OCR studies revealed impaired mitochondrial bioenergetics in *reep1* larvae at 120 hpf compared with control larvae at the same stage of development, with significant reductions in ATP production and maximal respiration (Figure 5F,F’), data similar to those observed in SPG31 cultured skin fibroblasts.

### 2.4. In Vitro and In Vivo Modulation of ROS Overproduction by Resveratrol

In order to explore the possible consequences of *REEP1* mutations in the pathogenesis of SPG31 disease, we evaluated the levels of oxidative stress in both primary fibroblasts and morphants. The cell-permeant indicator for reactive oxygen species (ROS) 2′,7′-dichlorodihydrofluorescein diacetate (H2DCFDA) was used to measure the degree of oxidation into its fluorescent form 2′,7′-dichlorofluorescein (DCF).

Fluorescence measurements showed a significant increase in susceptibility to oxidative stress in both human cells and zebrafish. To examine the cellular response to an oxidative insult, we exposed the fibroblasts to hydrogen peroxide, obtaining a six-fold increase in ROS levels. However, significant differences between mutated fibroblasts compared to WT ones were maintained even upon H_2_O_2_ exposition. 

To test the effectiveness of the antioxidant agent resveratrol to rescue ROS overproduction, we treated both cells (using a 25 µM concentration) and zebrafish (5 µM) for 24 h. Our data demonstrated the ability of resveratrol to restore normal cellular ROS levels, particularly during a stress condition. In the treated zebrafish model, resveratrol reduced fluorescence levels of oxidative stress with rescue to control level (Figure 6B).

## 3. Discussion

Whilst recessive mutations in *REEP1* are extremely rare, being observed in a single kindred with manifestations of upper and lower motor neuron involvement [34], the vast majority of variants in *REEP1* are AD-inherited and associated with pure degeneration of the corticospinal tract. Nevertheless, the mechanisms through which mutations lead to spastic paraplegia and SPG31 are poorly understood. Here, we confirmed that REEP1 is required for normal mitochondrial function and morphology, expanding its role in the regulation of redox homeostasis.

In SPG31 primary cell lines, we observed low mitochondrial ATP levels and increased sensitivity to oxidative stress; a similar pattern has been described in the cells of patients carrying biallelic mutations in paraplegin, a mitochondrial chaperone protein associated with HSP type 7 [35]. Moreover, loss of REEP1 function alters the balance between fusion and fission, through disruption of DRP1 function, with ensuing bioenergetic and oxidative defects [11]. We detected a significant increase in hyperfused and branched mitochondria, confirming that REEP1 and DRP1 work in similar pathways or are closely related [11]. Several lines of experimental evidence show an alteration in the process of DRP1 translocation to the mitochondria to promote fission [36] with a variation of basal ∆ψm and corroborating the main phenotypic findings in the SPG31 model [11,37]. In accordance with DRP1 null cells [38] and DRP1-mutated HeLa cells [33], we confirmed that FCCP-induced depolarization did not cause mitochondrial fragmentation in REEP1-deficient fibroblasts, indicating that mutual interaction plays a role in mitochondrial dynamics. These mechanisms have also been proposed to underlie different neurodegenerative diseases, such as motor neuron or peripheral neuropathies, in addition to other forms of HSP [39,40,41]. 

Muscle weakness and deficits in locomotor activity are prominent symptoms in affected patients, and mouse models, manifesting motor defects with denervation of neuromuscular junctions [42]. Our zebrafish *reep1* model showed impaired locomotion at different stages of development, mimicking the phenotype observed in murine *Reep1* models [37,42]. Moreover, spinal motor axons of the morphants exhibited shorter axonal length, suggesting that *ree*p1 may have an important role in motor axon outgrowth during the development, at least in zebrafish. Zebrafish KD experiments also demonstrated that zf-reep1 has an important role in oxidative stress regulation, a finding consistent with the results seen in SPG31 primary skin cells. Altogether, findings seen in zebrafish support the idea that *reep1* morphants recapitulate several phenotypes seen in *SPG31*-associated HSP, enabling the use of this model for further evaluation in vivo of *reep1* functions, especially in early brain development, a task easier to perform than in mice.

In our experimental model, we hypothesized a downregulation in the mitochondrial antioxidant systems as an effect of ROS scavenging. Oxidative stress seems to be one of the main inducers of neurodegeneration, altering mitochondrial membrane permeability and structure, membrane potential, and respiratory chain [43]. Bioenergetic defects, as well as aberrant mitochondrial morphology seen in SPG31 cells, lead to enhanced ROS formation, which may deteriorate mitochondrial health and further exacerbate oxidative stress. An increase in DRP1 phosphorylation correlates with the observed increase in ROS levels [44]. The collapse of the reticular form of mitochondria into fragments facilitates the mitophagy process, which selectively targets the dysfunctional mitochondria and promotes their removal [44,45]. In this scenario, intracellular accumulation of impaired mitochondria continues the toxic effect ROS-mediated able to impact the integrity of cellular macromolecules, such as mitochondrial DNA, nuclear DNA, and lipids, along with energy depletion and a local imbalance of calcium homeostasis [46], resulting in neuronal degeneration.

The use of antioxidants represents an effective strategy to restore the imbalance provoked by the excess level of ROS. Several endogenous and non-endogenous antioxidant agents have been widely described in the literature to play a role in halting free oxygen radicals toward redox balance, also in the context of neurodegeneration [47]. Resveratrol is one of the most convincing polyphenolic composites, which have the potential to scavenge ROS, induce antioxidant enzymes, and inhibit pro-oxidant pathways [48]. Evidence showed that it possesses strong antioxidant activity, and promising therapeutic properties have been described in several pathological conditions, including neurodegeneration [49,50]. Resveratrol treatment modulates specific signaling pathways, driving the formation of new healthy mitochondria and to the proper cAMP and MAM protein levels [51]. In our study, the use of resveratrol restored oxidative balance, ameliorating SPG31 cellular and improving the motor phenotypes of morphants.

In conclusion, our study has outlined the impact of mitochondrial metabolism in the pathophysiology of REEP1 deficiency and reinforced our understanding of an interplay between mitochondrial impairment and oxidative stress in SPG31. Increased susceptibility to oxidative stress plays a central role in triggering the neurodegeneration mechanism and might represent a valuable target to prompt new therapeutic approaches to counteracting disease progression. Additional investigations on alterations of the mitochondrial compartment in neuronal cells could reveal new insights into the cell pathology overcoming the limitations of cultured skin fibroblasts as a disease model. Furthermore, future studies have to be set to evaluate the ensuing consequences of both acute and chronic resveratrol treatments on the mitochondrial-related features in REEP1 disease.

## 4. Materials and Methods

### 4.1. Molecular Analyses

After obtaining patients’ informed consent, genetic analyses and skin biopsies were performed. Genomic DNA was extracted from the peripheral blood of patients of interest using the MagNA Pure Compact System (Roche Diagnostics, Monza, Italy). Targeted mutation analysis was performed by Sanger sequencing. Sequencing was performed in an ABI 3500 genetic analyzer (Applied Biosystems, Foster City, CA, USA) using ABI BigDye 3.1 chemistry, and traces were analyzed with SeqScape software (https://www3.appliedbiosystems.com accessed on 26 January 2023). Total RNA was isolated from skin fibroblasts obtained from punch biopsies in P1 and P2. This was performed using a High Pure RNA Isolation Kit (Roche Diagnostics GmbH, Mannheim, Germany), and the cDNA products were used directly to co-amplify the housekeeping *GAPDH* and *REEP1* transcripts.

### 4.2. Cell Culture

Patients (or their parents) signed an informed consent form for skin biopsy, authorizing research purposes in accordance with our Tuscany Region Ethics committee. Primary fibroblast cell lines were grown at 37 °C with 5% CO_2_ in Dulbecco’s modified Eagle’s medium (DMEM), containing 10% fetal bovine serum (FBS), 4.5 g/L glucose and 1% antibiotics/antimycotics. All cell lines were tested for mycoplasma contamination.

### 4.3. Western Blotting

For Western blotting, samples were homogenized in RIPA buffer (150 mM NaCl, 50 mM Tris-HCl, 6 mM EDTA, 1% NP-40, 0.1% SDS, 0.5% deoxycholic acid, pH 8.0) containing inhibitors of proteases (Roche Diagnostics GmbH, Mannheim, Germany) and centrifuged for 10 min at 14,000× *g* at 4 °C. In all, 15–50 μg of protein lysates, determined by BCA assay (Invitrogen-ThermoFisher Scientific, Waltham, MA, USA) was denatured and separated by electrophoresis using 8–16% Tris-Glycine Mini Gels (Invitrogen-ThermoFisher Scientific, Waltham, MA, USA) and then electro-blotted onto PVDF membranes (Bio-Rad Laboratories Inc., Hercules, CA, USA). Membranes were blocked with TBS/0.1%-Tween20 (TTBS) containing 5% non-fat dry milk before overnight incubation with the specified antibodies. Peroxidase-conjugated anti-mouse and anti-rabbit secondary antibodies (Jackson ImmunoResearch, Laboratories Inc., Cambridge, UK) were added for 1 h at room temperature in the same buffer as used for the primary antibodies (2.5% non-fat dry milk in TTBS). Reactive bands were detected using Clarity Max^TM^ Western ECL Substrate (Bio-Rad Laboratories Inc., Hercules, CA, USA), according to the manufacturer’s instructions. Densitometry of Western blot bands was performed with the ImageJ software. Primary antibodies used for Western blotting analysis were as follows: REEP1 (Sigma-Aldrich, St. Louis, MO, USA, #SAB2101976; dilution 1:1000); total DRP1 (BD Transduction Laboratories, Oxford, UK, #611112; dilution 1:500); DRP1 Ser 637 (Byorbyt #orb127984; dilution 1:500). Immunodetection with porin antibody (MitoSciences, Eugene, OR, USA, #MSA05; dilution 1:5000) served as a loading control to normalize the bands intensity.

### 4.4. Immunohistochemistry

For immunohistochemistry, embryos were treated with 0.005% phenylthiourea from 24 hpf to prevent the development of pigmentation. Whole-mount immunohistochemistry was performed in 48 or 120 hpf embryos fixed in 4% PFA overnight at 4 °C and stored in methanol as described in [52]. The antibodies used were mouse anti-Znp1 (ab113545, Abcam, 1:200 dilution); mouse anti-Acetylated-Tubulin (Life Technology, Carlsbad, CA, USA, 018M4788V, 1:500 dilution) [25]. Mitochondrial respiration and oxygen consumption rate (OCR) were performed using the XF24 extracellular flux analyzer (Seahorse Bioscience, Billerica, MA, USA) as reported in [53].

### 4.5. Treatments

Both primary fibroblasts and the zebrafish model were treated with Resveratrol (R5010, Sigma-Aldrich, St. Louis, MO, USA) dissolved in DMSO. Stock solutions were further diluted in regular cell culture medium or zebrafish E3 medium to the final concentration of 25 µM and 5 µM, respectively. Rescue experiments were performed following 24 h of treatment in regular medium in the presence/absence of resveratrol. The used concentrations are based on the results from previous experiments and literature findings and were chosen for the absence of cytotoxicity up to 48 h of treatment [51]. Twenty-four hours of EBSS-induced starvation was assessed to activate the cAMP-dependent protein kinase to force DRP1 phosphorylation in primary fibroblasts. To mimic an oxidative insult, cells were exposed for 30 min in the presence/absence of hydrogen peroxide at 500 µM.

### 4.6. Analysis of Mitochondrial Network

Mitochondrial morphology was assessed by staining cells with 10 nM MitoTracker Red (Invitrogen, Carlsbad, CA, USA) for 30 min at 37 °C. Cellular fluorescence images were acquired using a Nikon Ti2-E inverted microscope equipped with a DS-Qi2Mc camera and collected with a Nikon ×60 Plan Apocr λ (NA = 1.40) oil immersion objective, using a TRITC filter set. Mitochondrial networks were analyzed, outlining three different morphologies of mitochondrial compartment: Group 1: circularity 0–0.3 consisted of tubular mitochondria; Group 2: circularity 0.3–0.6 consisted of intermediate mitochondria; Group 3: circularity 0.6–1 consisted of fragmented mitochondria. For the image analysis, we also considered the mitochondrial AR, which is a function of mitochondrial length; and the mitochondrial FF (perimeter 2/4π·area), which is a combined measure of both mitochondrial length and degree of branching. Both AR and FF are independent of image magnification [54]. For each experiment, at least 20 cells derived from REEP1 fibroblasts and controls were used to calculate each of the different mitochondrial population areas as a percentage of the total mitochondrial compartment area. Data analysis was performed using the “Mitochondrial Morphology” macro [55] in ImageJ (http://rsbweb.nih.gov/ij/, accessed on 12 September 2022).

### 4.7. Evaluation of Mitochondrial Membrane Potential (ΔΨm)

Fibroblasts were plated at 1.5 ×10^4^ cells/well density in 96-well plates with normal growth medium. Following 24 h of growth, the mitochondrial membrane potential was measured using the fluorescent dye tetramethylrhodamine methyl ester (TMRM, Invitrogen™, Carlsbad, CA, USA). The dye was loaded into cells in 100 nM in bicarbonate and phenol red-free Hank’s balanced salt solution (HBSS) supplemented with 10 mM HEPES (Sigma-Aldrich, St. Louis, MO, USA), 2 µM cyclosporine-H (CsH), pH 7.4 and placed at 37 °C for 5 min. Fluorescence was measured on a SpectraMax^®^ ID3 plate reader (Molecular Devices, San Jose, CA, USA) (544/590 nm Ex/Em, bottom reading). The assay was performed in parallel as described above, with the addition of 20 μM FCCP, which collapses the mitochondrial membrane potential. All data were expressed as the total TMRM relative fluorescence units (RFU) minus the FCCP-treated TMRM fluorescence and normalized to the number of cells using 4′,6-diamidino-2-phenylindole (DAPI) staining (358/461 nm Ex/Em, bottom reading). Kinetic evaluation of ΔΨm was performed by live imaging as previously reported [56]. Briefly, fibroblasts were seeded at 60% confluence on 35 mm glass bottom dishes (WillCo Wells B.V., Amsterdam, The Netherlands) and grown for two days in DMEM. Cells were incubated in bicarbonate and phenol red-free HBSS, supplemented with 10 mM HEPES (Sigma-Aldrich, St. Louis, MO, USA) and 1.6 µM CsH, and loaded with 20 nM TMRM for 30 min at 37 °C. Cellular fluorescence images were acquired every minute using a Nikon Ti2-E inverted microscope equipped with a DS-Qi2Mc camera and collected with a Nikon ×60 Plan Apocr λ (NA = 1.40) oil immersion objective, using a TRITC filter set. After 7 min of baseline, oligomycin (2.5 µM final concentration) was added to the media, recording sequential digital images for 15 min. At the end of each experiment, mitochondria were fully depolarized by the addition of 4 µM FCCP. Clusters of mitochondria (15–30 on average) were identified as regions of interest (ROIs), and fields without cells were used as a background. In all sequential digital images and for each ROI, the changes in fluorescence intensity were measured using ImageJ software. Fluorescence values were expressed as a percentage of the controls’ baseline (*T*_0_, 100%) and reported as the average ROIs ± SD for each time point.

### 4.8. Oxygen Consumption Rate Measurements

Oxygen consumption rate (OCR) was measured in REEP1 cultured fibroblasts and zebrafish (together with their respective controls) using an XFe24 Extracellular Flux Analyzer (Seahorse Bioscience, Agilent, Santa Clara, CA, USA). Cells were plated in XF 24-well cell culture microplates at 5 ×10^4^ cells/well, whereas embryos were captured on an islet capture plate at 120 hpf in accordance with [54]. Plates preparation and injection strategy adopted for OCR measurements followed the standard procedure for the *Mito Stress Test* as previously reported [57,58]. Data were expressed as pmol of O_2_/min normalized post-assay by the fluorescence CyQUANT Cell Proliferation Assays (Invitrogen™, Carlsbad, CA, USA), as reported by the producer’s guidelines [59].

### 4.9. Functional Studies in Zebrafish

Adult male and female wild-type (WT) were housed according to standard procedures on a 14-hour light/10-hour dark cycle [60]. Zebrafish embryos and larvae were collected and raised at 28.5 °C in E3 medium using established procedures and staged in hours post fertilization (hpf) or days post fertilization (dpf) [61,62]. All experiments were conducted in accordance with the European Union (EU) Directive 2010/63/EU for animal experiments, and under the supervision of the Institutional Animal Care and Use Committee (IACUC) of the University of Pisa and following the 3Rs principles [27,63]. The zebrafish *reep1* gene, which maps to chromosome 17, consists of 8 coding exons and encodes a 308 amino acid receptor accessory protein 1. Multiple alignments of REEP1 amino acid sequences were performed using Protein BLAST (https://blast.ncbi.nlm.nih.gov/ accessed on 26 January 2023) (see Appendix A). This in silico analysis revealed a high conservation of reep1 with a 74% of identity between the human (ENST00000538924.7) and *zebrafish* (ENSDARG00000014854**)** protein sequence, suggesting that reep1 acts as the fish ortholog of the human counterpart with a conserved putative function.

Total RNA was extracted from 30 embryos at different stages, as reported in the text, using Quick RNA miniprep (Zymo Research, Irvine, CA, USA) according to the manufacturer’s instructions. cDNA and qRT-PCR were performed as described in [64,65]. Relative expression levels of each gene were calculated using the 2^−ΔΔCt^ method [66,67]. The results obtained in at least three independent experiments were normalized to the expression of the housekeeping gene, β-actin (ENSDARG00000037746). The mean of the controls was set**.** Primers for mRNA sequences were designed using the zebrafish sequence of *reep1* (ENSDARG00000014854). Antisense and sense riboprobe synthesis was performed using the Digoxigenin (DIG) RNA Labelling Kit (Roche, Basel, Switzerland), and WISH was performed [26,60]. For the generation of zebrafish *reep1* KD, we used *reep1* antisense Morpholino oligonucleotides (MO) provided by Gene Tools, Philomath, OR, USA. MO ATG *reep1* sequence: 5′ATAATCCAGGAGACCATTGCGCTGT-3′. The concentration of MO was carefully titrated to avoid nonspecific binding effects to assess specificity to *reep1*. After titration, we used, in all experiments, 10 ng/nl of MO ATG *reep1* against *reep1*. Rescue experiments were performed through co-injection of 300 pg of the open reading frame of human *REEP1* (REEP1 (NM_022912) Human Tagged ORF Clone, Origene) subcloned into the pCS2+. Each experiment was repeated at least three times if not otherwise stated. Touch-evoked escape response was measured at 48 hpf on a semi-quantitative scale from normal, as reported in [68]. Coiling behavior was measured in embryos at 30 hpf (*n*= 30 for each experiment) using the Danioscope software (Noldus©, Wageningen, The Netherlands). We also analyzed the locomotion in 120 hpf larvae for each experimental group. The larvae were transferred into 96 multi-well plates containing 300 μL of egg water per well. The plate was placed in the DanioVision^®^ device (Noldus©, Wageningen, The Netherlands), and the larval activity was recorded for 30 min and analyzed through EthoVision XT^®^ software (Noldus©, Wageningen, The Netherlands). Statistical analysis was performed considering three independent biological replicate experiments, and the data were plotted as the mean ± standard error of the means.

### 4.10. Determination of Reactive Oxygen Species

For the evaluation of intracellular reactive oxygen species (ROS) production, the in vivo carboxy-H2DCFDA fluorescent probe (#8206004, Abcam, Cambridge, MA, USA) was used. Skin fibroblasts were labeled at 25 μM for 45 min at 37 °C and then cultured for an additional hour in the presence/absence of hydrogen peroxide at 500 nM to mimic ROS stimulus. REEP1 cells were analyzed on a SpectraMax^®^ ID3 plate reader (Molecular Devices, San Jose, CA, USA) at wavelengths Ex/Em: 485/535 nm, and the difference in ROS levels between treated and untreated conditions were expressed as relative fluorescent units (RFU) after background subtraction. DCF signal inside wells was normalized to Hoechst 33342 intensity which is dependent on the cell number.

Zebrafish embryos at 24 hpf were incubated with 30 µM of carboxy-H2DCFDA fluorescent probe for 40 min in the dark and then washed three times with E3 medium, as reported [69]. A lateral image of each larva was acquired using a fluorescence microscope, and the fluorescence intensity in the selected ROI was quantified using ImageJ software. Data were normalized to background fluorescence.

A comparative analysis was performed both in cells and zebrafish models, treated with resveratrol (see treatments section) to evaluate the rescue from ROS overproduction disease-related, anticipating new possible options for therapy.

### 4.11. Statistical Analyses

All data in the manuscript represent two or more independent experiments giving similar results. Normality tests by GraphPad Prim 9 software were performed to verify the Gaussian distribution of the data. The significance between various groups/treatments was determined by parametric or non-parametric tests. For each experiment, a specific statistic method was proposed together with the sample size parameter. Statistical significance was indicated as * *p* < 0.05, ** *p* < 0.01, *** *p* < 0.001, and **** *p* < 0.0001.

## Figures and Tables

**Figure 1 ijms-24-03527-f001:**
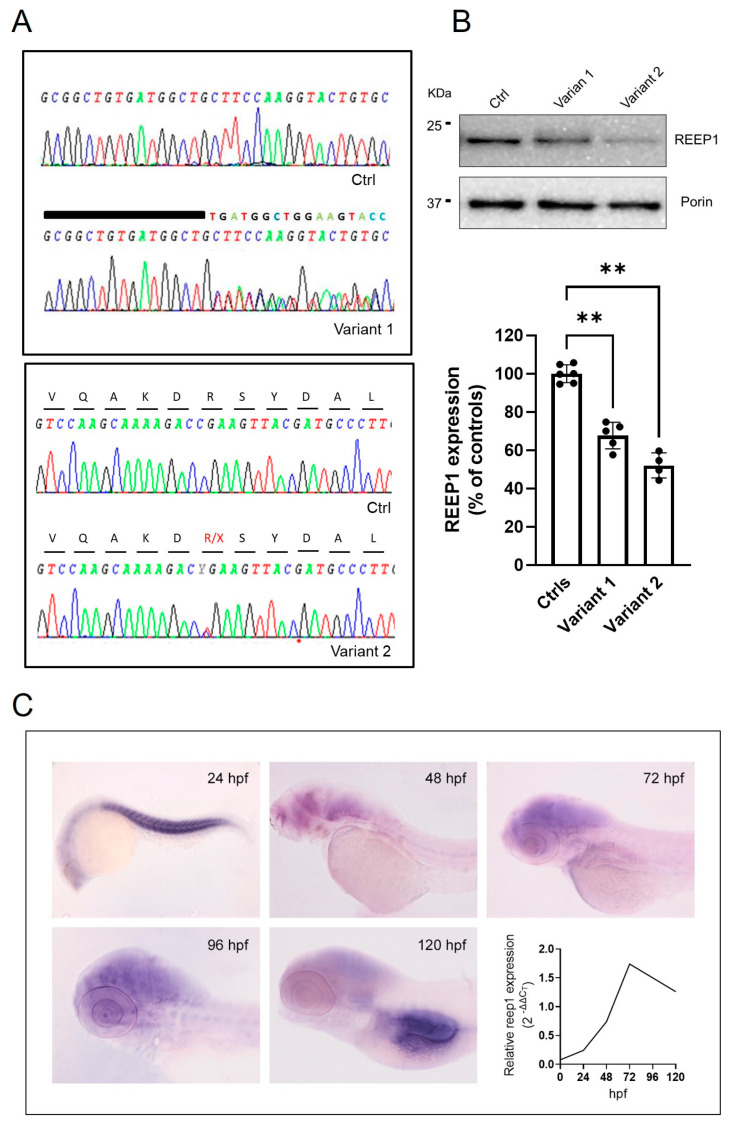
Molecular studies in the analyzed disease models. (**A**) Genetic studies of DNA from mutated fibroblasts show the electropherogram of exon 5 flanking the c.401_409dup9del19 and c.337C > T mutations. A wild-type REEP1 sequence is also reported as a reference. (**B**) Representative WB and quantification of REEP1 protein expression levels related to both mutant cell lines and control. Porin expression was used as a loading control. A significant reduction in REEP1 protein expression was observed with residual levels between 50 and 70% compared to controls. Three different control fibroblasts (in technical duplicate) and two subjects for each mutation (in technical triplicate) were analyzed and plotted both as individual values and histograms with mean and SD. ANOVA test (one-way ANOVA) was used to compare the mean levels of each mutated line with the control mean. ** *p* < 0.01. (**C**) Spatial-temporal expression of reep1 mRNA during zebrafish embryogenesis. WISH of zebrafish reep1 was performed at different developmental stages (from 24 to 120 hpf), and qRT−PCR analysis showed reep1 developmental relative expression from 0 hpf to 120 hpf in WT zebrafish.

**Figure 2 ijms-24-03527-f002:**
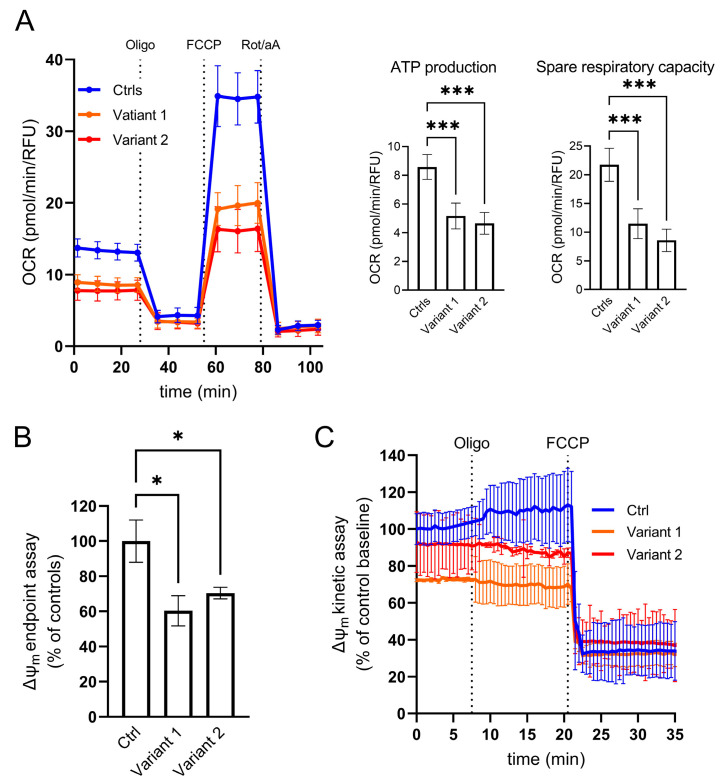
Cellular analysis of mitochondrial function. (**A**) Micro-oxygraphy track showing a reduced OCR in mutated cell lines, which is clearly evident after FCCP injection, reflecting a deficient spare respiratory capacity. Data represent mean ± SEM of controls (*n* = 3) and patients (*n* = 2 for each mutation). Two independent experiments were run with four technical replicates for each cell line. (**B**–**C**) Mitochondrial membrane potential profiles evaluated both in terms of TMRM probe accumulation and membrane potential maintenance. (**B**) End-point assay indicating a mitochondrial membrane depolarization in mutated fibroblasts reported as percentage of controls. Data were normalized by DAPI staining as a function of cell number. (**C**) Kinetic track demonstrates the ability of REEP1 cultured skin fibroblasts to maintain polarized mitochondrial membrane after oligomycin blocking proton transit through Complex V, highlighting any leakage of the inner mitochondrial membrane. FCCP was added at the end of the experiments to fully depolarized mitochondrial to demonstrate the specificity of measurements. Data represent mean ± SEM of controls (*n* = 3) and patients (*n* = 2 for each mutation). A technical triplicate was assessed for endpoint assay, whereas for kinetic analyses, cell lines were acquired in an independent experiment recording 10 different ROIs/field over time. For all the reported experiments, statistics were assessed by ordinary ANOVA test (one-way ANOVA). * *p* < 0.05; *** *p* < 0.001.

**Figure 3 ijms-24-03527-f003:**
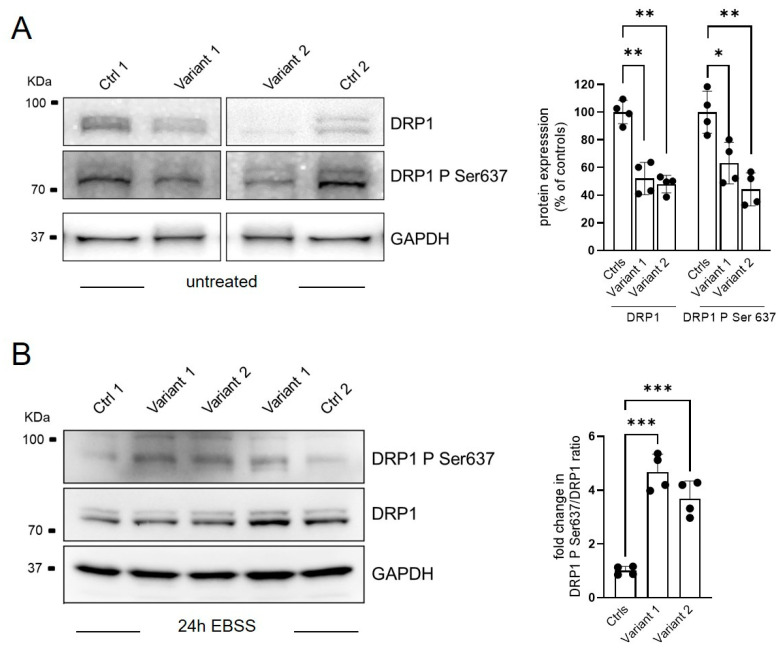
Regulation of mitochondrial dynamics via DRP1-Ser637 phosphorylation status. Representative WB analysis of DRP1 phosphorylation process in basal condition (**A**) and after 24 of starvation (EBSS treatment) (**B**). Two different control fibroblasts and two subjects for each mutation in technical duplicate were analyzed and plotted as histograms with both mean ± SD and individual values of replicates. Ordinary ANOVA test (one-way ANOVA) was used to compare the mean levels of each mutated line with the control mean. * *p* < 0.05; ** *p* < 0.01; *** *p* < 0.001.

**Figure 4 ijms-24-03527-f004:**
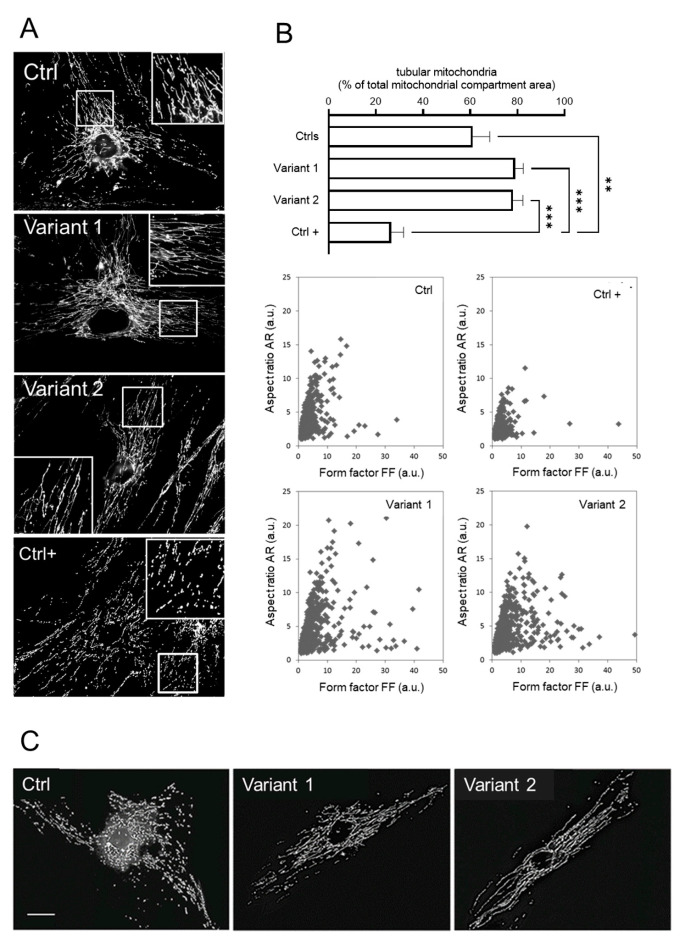
Mitochondrial morphology investigations in skin fibroblasts. (**A**) The mitochondrial network analysis by fluorescence microscopy showed hyperfused tubules in REEP1 mutated fibroblasts as compared with control. A control line under the effect of FCCP was used as positive control of the fragmented network (Ctrl +). (**B**) Morphometric analysis showing a significant increase in tubular mitochondria in patients. Statistics was assessed by ordinary ANOVA test (one-way ANOVA). ** *p* < 0.01; *** *p* < 0.001. Computer-assisted morphological analyses of mitochondrial AR plotted as function of FF showed in SPG31 patients’ cells higher values for both FF and AR (tubular and hyperfused mitochondria). Ctrl + indicates a positive control with very low levels of fragmented mitochondria. (**C**) Representative fluorescence microscopy images of MitoTracker Red in fibroblasts 30 min after treatment with 2 µM FCCP. Unlike the control line, no fragmented mitochondria were observed in both mutated fibroblasts. Scale bar: 20 µm.

**Figure 5 ijms-24-03527-f005:**
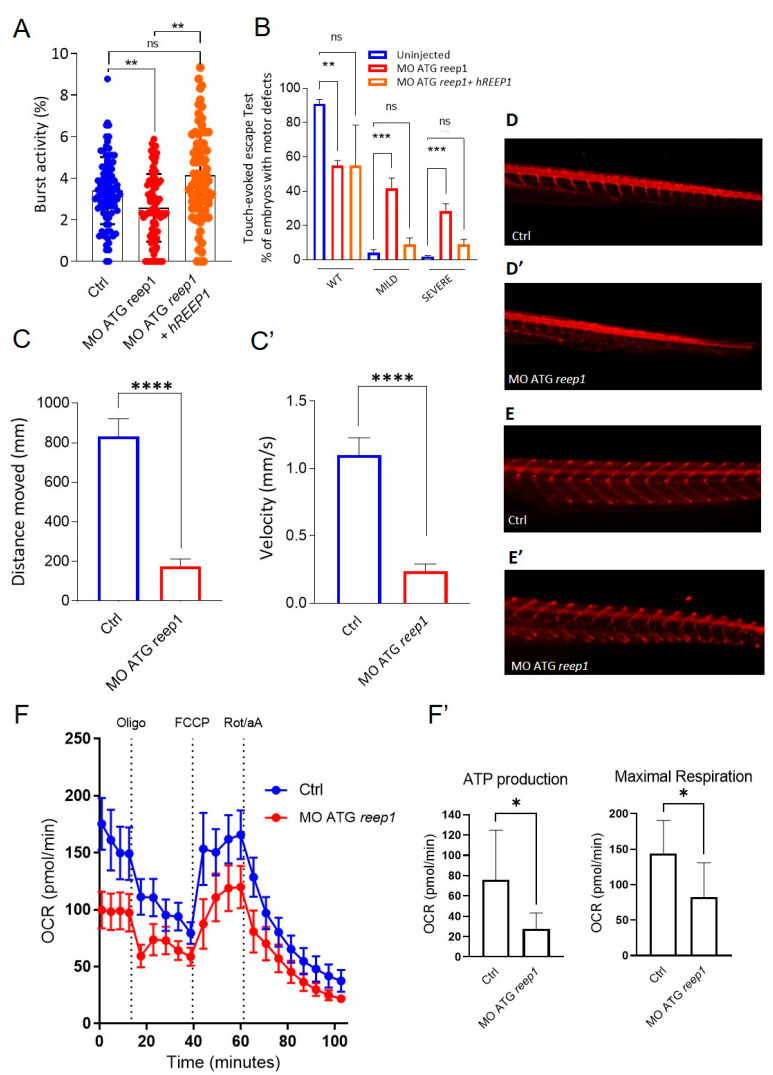
Phenotype characterization of the reep1 zebrafish model. (**A**) Coiling frequency in zebrafish embryos at 30 hpf is decreased in reep1 morphant compared with WT (MO ATG *reep1 n* = 90; controls *n* = 90; MO ATG *reep1* + h*REEP1 n* = 90, in 3 independent experiments). The co-injection of embryos at one-cell stage with human mRNA/REEP1 and reep1 MO ATG was able to rescue the locomotor impairment. (**B**) Touched evoked response analysis, showing decreased movement in response to touch stimulus of morphants at 48 hpf compared to uninjected controls. The co-injection of embryos at 48 hpf with human mRNA/*REEP1* and reep1 MO ATG was able to rescue the touched evoked phenotype. Statistics were calculated by a two-tailed unpaired *t*-test. (**C**,**C’**). Automated analysis of spontaneous motor activity revealed a reduction in swim distance and velocity in *reep1* morphant larvae at 120 hpf compared to control siblings (MO ATG reep1 *n* = 120; controls *n* = 120, in 3 independent experiments). (**D**,**D’**). Lateral tail views at 48 hpf of whole-mount larvae labeled with anti-acetylated alfa-tubulin (MO ATG *reep1 n* = 40; controls *n* = 40). (**E**,**E’**) Lateral tail views at 120 hpf of whole-mount larvae labeled with anti-znp1 (MO ATG reep1 *n* = 40; controls *n* = 40). (**F**,**F’**) Mitochondrial respiratory analysis of controls (*n* = 20) and *reep1* morphant larvae (*n* = 20) at 120 hpf. Statistics were calculated by Mann–Whitney test in panels (**A**,**C**,**F**), whereas in panel (**B**) by two-tailed unpaired *t*-test. * *p* < 0.05, ** *p* < 0.01, ****p* < 0.001; **** *p* < 0.0001, ns, not significant.

**Figure 6 ijms-24-03527-f006:**
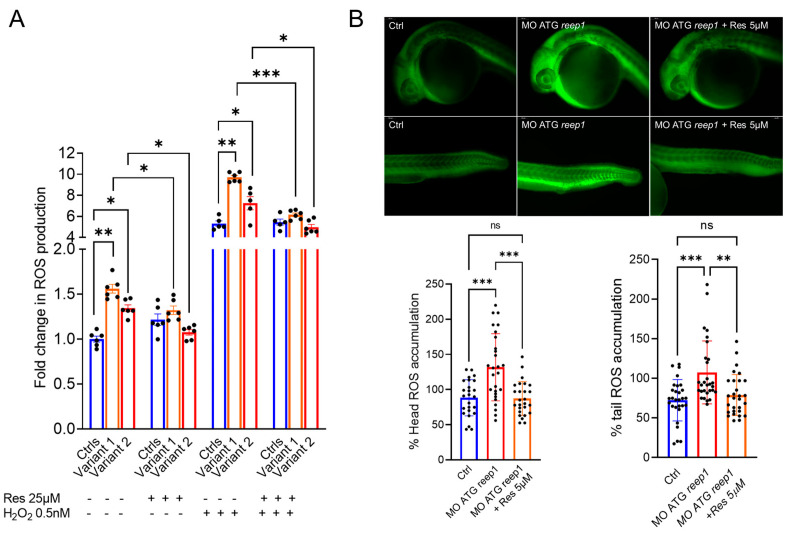
Evaluation of the susceptibility to oxidative stress in REEP1/SPG31 disease models and resveratrol modulatory effect on ROS overproduction. (**A**) Under both regular medium (RM) and stress conditions, patients produced a significantly greater amount of ROS compared with controls, indicating an increased susceptibility to oxidative stress. Resveratrol treatment reduces the ROS levels restoring the control values. Data represent mean ± SEM of controls (*n* = 3) and patients (*n* = 2 for each mutation) analyzed in technical duplicate (controls) or triplicate (patients). Statistical analysis was performed by ordinary ANOVA test (one-way ANOVA). * *p* < 0.05; ** *p* < 0.01; *** *p* < 0.001. (**B**) Representative fluorescence images of ROS generation in zebrafish larvae at 24 hpf (MO ATG *reep1 n* = 30; controls *n* = 30; MO ATG *reep1*+ Res 5µM *n* = 30, in 3 independent experiments. Graphs show the quantitative analysis of fluorescent signals. *** *p* < 0.01 was calculated by Dunnett’s multiple comparisons test. Abbreviations: *n*, number of evaluated embryos in total; ns, not significant. The values are expressed as mean ± standard deviation (SD).

## Data Availability

All data generated or analyzed during this study are included in this published article and its Supplementary Information Files.

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
