# Peer review of "Converging Role for REEP1/SPG31 in Oxidative Stress"

_ijms, 2023, doi:10.3390/ijms24043527_

Round 1

Reviewer 1 Report

It is a well-presented manuscript with a complete experimental design that meets the proposed objectives. During the reading of it I have observed some minor errors that could be corrected before its publication (errors more in format than in content)

Author Response

We wish to express our appreciation for the Reviewer for providing such positive comments on the manuscript. We have carefully proof-read the text and properly corrected grammatical, syntax and typing errors.

Reviewer 2 Report

In the article “Convergent role of REEP1/SPG31 in oxidative stress”, a multi-stage experimental analysis was performed demonstrating effects similar to human and mammalian paraplegia in the reep1 model in zebrafish. In particular, the authors showed impaired locomotion at different developmental stages mimicking the phenotype seen in mouse models of Reep1.

In general, the results of this work are worthy of publication, but it is necessary to make corrections and clarify some points.

 During the review process, the following questions and comments to the authors of this work arose.

 Major:

1. In the Introduction section, the problem of this study is not fully disclosed. It is necessary to additionally formulate a hypothesis, which is tested experimentally in this work.

2. The use of the zebrafish model for the manifestation of the effects of these genes looks convincing and very effective. However, it is necessary to further clarify how similar the effects of genes are in humans and zebrafish.

 3. Figure 2 indicates that the data represent the Mean ± Standard Error of the mean for controls (n=3) and patients (n=2 for each mutation). Two independent experiments were performed with four technical replicates for each cell line. How does the specified number of experiments ensure reliability? It is necessary to more clearly justify the number of experiments and replicates performed, indicating the level of confidence that provides statistically significant data.

 4. The authors indicate that reduced DRP1 expression was observed by WB also in our cells (Figure 3A), while sustained levels of OPA1, MNF2, and FIS1, additional proteins involved in mitochondrial dynamics, were similar to those observed in control fibroblasts (data not shown) . It is necessary to bring this data into the saplimentary file.

 5. In Fig. 4.A, the authors indicate that the control line under the influence of FCCP was used as a positive control of the fragmented network (Ctrl +). Morphological differences in Fig. 4A is not visible. What are these differences?

  6. In the Statistical analysis section, you need to add information justifying the use of parametric or non-parametric tests used in the work.

Author Response

We wish to express our appreciation for the Reviewer for providing such positive comments on the manuscript, as well as for the accuracy and constructiveness with which He/She expressed the specific comments.

A detailed point- by -point reply has been proposed below:

  1. In the Introduction section, the problem of this study is not fully disclosed. It is necessary to additionally formulate a hypothesis, which is tested experimentally in this work.

We thank the expert referee for his/her suggestion. We've slightly edited the concluding part of the introduction (lines 75-89), to provide a clearer message

  1. The use of the zebrafish model for the manifestation of the effects of these genes looks convincing and very effective. However, it is necessary to further clarify how similar the effects of genes are in humans and zebrafish.

We thank the expert referee for his/her precious suggestion. A dedicated paragraph has been added in the M&M section describing how the evolutionary conservation of reep1 sequence in vertebrates was verified. (lines 517-525). Although “humanizing” fish data is always thorny, reep1 morphant zebrafish model replicates the main “clinical” features seen in REEP1- related disease (see lines 335-340). These findings make the zebrafish a compelling experimental tool for the study of genotype/phenotype correlations, thanks to the presence of a nervous system organization very similar to that of human.

  1. Figure 2 indicates that the data represent the Mean ± Standard Error of the mean for controls (n=3) and patients (n=2 for each mutation). Two independent experiments were performed with four technical replicates for each cell line. How does the specified number of experiments ensure reliability? It is necessary to more clearly justify the number of experiments and replicates performed, indicating the level of confidence that provides statistically significant data.

Although in our case the sample size is closely connected to the number of the available cell lines, level of reproducibility was set using a confidence interval at 95%. If the reviewer deems it necessary, we can provide new graphs including the mean (or median) with 95% of CI

  1. The authors indicate that reduced DRP1 expression was observed by WB also in our cells (Figure 3A), while sustained levels of OPA1, MNF2, and FIS1, additional proteins involved in mitochondrial dynamics, were similar to those observed in control fibroblasts (data not shown). It is necessary to bring this data into the supplementary file.

Following the Reviewers’ suggestion, we provided a supplementary figure showing the steady-state level of the additional proteins involved in mitochondrial dynamics.

  1. In Fig. 4.A, the authors indicate that the control line under the influence of FCCP was used as a positive control of the fragmented network (Ctrl +). Morphological differences in Fig. 4A is not visible. What are these differences?

Following the Reviewers’ comment we added magnifying inserts inside the new Figure 4A to better show the morphological differences in mitochondrial network as quantified in figure 4B. Positive control show a low percentage of a tubular mitochondria due to the action of the uncoupler FCCP which induce a marked network fragmentation.

  1. In the Statistical analysis section, you need to add information justifying the use of parametric or non-parametric tests used in the work.

We thank the expert referee. We added the information justifying the use of parametric or non-parametric tests used in the work. We modified the text at lines 530-537.

Reviewer 3 Report

1. The English need improvement since there are some grammatical and syntax errors in the manuscript. For example,

·         in line number 19, the words “regulation” may be as “the regulation”;

·         in line number 48, “lipid droplets” as “in lipid droplet”;

·         in line number 71, “an inhibition” as “inhibition”;

·         in line number 82, “dramatic” as “a dramatic”;

·         in line number 89, “Supplementary” as “The supplementary”;

·         in line number 111, “reference” as “a reference”;

·         in line number 112, “a control” as “control”;

·         in line number 112, “loading” as “a loading”;

·         in line number 121, “mitochondrial” as “the mitochondrial”;

·         in line number 138, “TMRM” as “the TMRM”;

·         in line number 150, “inner” as “the inner”;

·         in line number 151, “specificity” as “the specificity”;

·         in line number 165, “EBSS” as “the EBSS”;

·         in line number 177, “as function” as “as a function”;

·         in line number 187, “resist to” as “resist”;

·         in line number 189, “depends to” as “depends on”;

·         in line number 194, “control” as “the control”;

·         in line number 198, “function” as “a function”;

·         in line number 206, “locomotor” as “the locomotor”;

·         in line number 218, “significantly” as “a significantly”;

·         in line number 240, “two-tailed” as “a two-tailed”;

·         in line number 263, “treated” as “the treated”;

·         in line number 293, “variation” as “a variation”;

·         in line number 294, “SPG31” as “the SPG31”;

·         in line number 311, “more easy” as “easier”;

·         in line number 318, “Increase” as “An increase”;

·         in line number 318, “of ROS” as “in ROS”;

·         in line number 319, “Collapse” as “The collapse”;

·         in line number 347, “peripheral” as “the peripheral”;

·         in line number 392, “zebrafish” as “the zebrafish”;

·         in line number 427, “Assay” as “The assay”;

·         in line number 428, “addition” as “the addition”;

·         in line number 446, “controls'” as “the controls'”;

·         in line number 446, “average of” as “the average”;

·         in line number 477, “Concentration” as “The concentration”;

·         in line number 498, “a wavelengths” as “wavelengths”;

·         in line number 501, “dependent to” as “dependent on”;

·         in line number 508, “with of” as “with”;

·         in line number 511, “Prims” as “Prism”.

The grammar mistakes which are not mentioned here are also to be checked and corrected properly.

3. There are some typing mistakes as well, and authors are advised to carefully proof-read the text. For example,

·         in line number 24, the word “knockeddown” may be as “knocked down or knocked-down”;

·         in line number 50, “key-site” as “key site”;

·         in line number 109, “showing” as “show”;

·         in line number 116 and 174, 212 “controls” as “control”;

·         in line number 122, “real time” as “real-time”;

·         in line number 135, “(Figure 2C) ,” as “(Figure 2C),”;

·         in line number 153, “end point” as “endpoint”;

·         in line number 161, “steady state” as “steady-state”;

·         in line number 194, “showing” as “showed”;

·         in line number 222, “pan neuronal” as “pan-neuronal”;

·         in line number 236, “cell-stage” as “cell stage”;

·         in line number 263, “stress-condition” as “stress condition”;

·         in line number 266, “disease-models” as “disease models”;

·         in line number 275, “com-parisons” as “comparisons”;

·         in line number 297, “play” as “plays”;

·         in line number 312, “models we” as “model we”;

·         in line number 312, “down regulation” as “downregulation”;

·         in line number 330, “composite” as “composites”;

·         in line number 338, “improved” as “improving”;

·         in line number 358, “Ethic” as “Ethics”;

·         in line number 499, “condition” as “conditions”.

The typos not mentioned here are also to be checked and corrected properly.

3. Check the abbreviations throughout the manuscript and introduce the abbreviation when the full word appears the first time in the abstract and the remaining for the text and then use only the abbreviation (For example, ROS, SOD, whole-mount in situ hybridization (WISH), aspect ratio (AR), etc.,). Make a word abbreviated in the article that is repeated at least three times in the text, not all words  to be abbreviated. The authors should avoid the usage of abbreviations in the keywords.

5. The superscript and subscript should be properly mentioned for the chemical names throughout the manuscript. For example, “CO2”.

6. The limitation of the present investigation may be given along with conclusion or under separate heading for understanding the concepts clearly.

Author Response

We wish to express our appreciation for the Reviewer for very thorough revision of English language, greatly contributing to readability of the submitted manuscript.

We have carefully revised the text rectifying grammatical, syntax and typing errors.

Moreover, we added a final paragraph in the discussion section (line 341-350) to better outline both strengths and limitation of the study.

Round 2

Reviewer 2 Report

After correction, this work can be published in IJMS